# Emergent constraints on future projections of the western North Pacific Subtropical High

Xiaolong Chen [1,2✉], Tianjun Zhou [1,2,3], Peili Wu [4], Zhun Guo[5] & Minghuai Wang[6]

The western North Pacific Subtropical High (WNPSH) is a key circulation system controlling the summer monsoon and typhoon activities over the western Pacific, but future projections of its changes remain hugely uncertain. Here we find two leading modes that account for nearly 80% intermodel spread in its future projection under a high emission scenario. They are linked to a cold-tongue-like bias in the central-eastern tropical Pacific and a warm bias beneath the marine stratocumulus, respectively. Observational constraints using sea surface temperature patterns reduce the uncertainties by 45% and indicate a robust intensification of the WNPSH due to suppressed warming in the western Pacific and enhanced land-sea thermal contrast, leading to 28% more rainfall projected in East China and 36% less rainfall in Southeast Asia than suggested by the multi-model mean. The intensification of the WNPSH implies more future monsoon rainfall and heatwaves but less typhoon landfalls over East Asia.

[1] State Key Laboratory of Numerical Modeling for Atmospheric Sciences and Geophysical Fluid Dynamics, Institute of Atmospheric Physics, Chinese Academy of Sciences, Beijing 100029, China. [2] CAS Center for Excellence in Tibetan Plateau Earth Sciences, Chinese Academy of Sciences (CAS), Beijing 100101, China. [3] University of the Chinese Academy of Sciences, Beijing 100049, China. [4] Met Office Hadley Centre, Exeter EX1 3PB, UK. [5] Climate Change Research Center, Institute of Atmospheric Physics, Chinese Academy of Sciences, Beijing 100029, China. [6] School of Atmospheric Sciences & Joint International Research Laboratory of Atmospheric and Earth System Sciences, Nanjing University, Nanjing 210023, China. ✉email: chenxiaolong@mail.iap.ac.cn

One-third of the world population live in East and Southeast Asia. Every year, people in this region suffer from many extreme weather and climate events such as floods, heatwaves, and typhoons, mostly resulting from anomalies of a dominant circulation system, the western North Pacific Subtropical High (WNPSH)[1–4]. The WNPSH is the western part of the North Pacific Subtropical High (NPSH) extending to the East Asian coast. On interannual to interdecadal timescales, stronger WNPSH usually causes more flooding and heatwaves in East Asia[5–7], while less tropical storms and typhoon landfalls[4,8]. Future changes of the WNPSH in the background of global warming have crucial implications to this densely populated and economically energetic region while remaining very uncertain, despite a number of focused studies in the past decade[9–13]. Under projected future warming, half of state-of-the-art climate models participating in the fifth phase of Coupled Model Inter-comparison Project (CMIP5) predict an intensification while the other half a weakening of the WNPSH[9]. Zonal oscillatory behavior of the WNPSH under the green-house gas (GHG) forcing[13] exacerbates the uncertainty in the future projection. Reducing uncertainty of future projection of the WNPSH is of paramount importance for policymakers and the society to prepare for climate change impact and adaptation.

Model errors in simulating the observed climate can degrade the reliability of regional projections[14]. Two well-known chronic biases in current climate models are a cold tongue bias in the equatorial Pacific[15–17] and the poorly simulated stratocumulus clouds over the tropical and subtropical oceans[18–22]. These biases in simulations of the historical climate can lead to uncertainties in the simulated surface warming patterns[23,24] and climate sensitivity under GHG forcing[25,26]. These are also potential sources of uncertainty for the projections of WNPSH as the WNPSH is strongly modulated by sea surface temperature (SST) anomalies (such as El Niño)[1,3,6,7,9] and land–sea thermal contrast[27–29].

Here we show that the SST biases in model simulations of the historical climate in the Pacific and Atlantic are closely linked with the intermodel spread of future WNPSH projections. These SST biases are manifested in two particular modes associated with the tropical cold tongue and marine stratocumulus errors, which can be observationally constrained. In a hierarchical statistical framework[30], these constraints based on relationships between future and current climate states (called emergent constraints[31]) can lead to a robust intensification of the WNPSH in a warmer future with about 45% of the projection uncertainty reduced.

## Results

**Leading modes of uncertainties in the WNPSH projection.** Mean-state changes of sea level pressure (SLP) during boreal summer (June, July, and August) between RCP8.5 scenario (2050–2099) and historical simulation (1956–2005) from 35 CMIP5 models ("Methods" and Supplementary Table 1) are used to represent future projections of the WNPSH. The largest uncertainty appears along the WNPSH ridge, extending from East China to the western North Pacific (WNP; 20–40°N, 110–160°E), where less than 70% of the models agree on the sign of change and the signal-to-noise ratio is below 0.5 (Supplementary Fig. 1). An empirical orthogonal function (EOF) analysis on the intermodel spread (see "Methods") has revealed the first two leading principal components (PC1 and PC2) account for nearly 80% of the intermodel variance (Supplementary Fig. 2). EOF1 shows a monopole structure with negative SLP anomalies stretching all the way from South Asia to the WNP, representing a weaker WNPSH (Fig. 1a). EOF2 shows a dipole structure, representing both the enhanced WNPSH and Asian Low and underlining the large-scale land–sea thermal contrast (Fig. 1b).

Because these modes are independent of models, it is possible to reduce the uncertainty of WNPSH projections if the PCs could be observationally constrained.

**Historical SST patterns related to the uncertainty modes.** By regressing historical mean state of SST patterns in boreal summer (June–July–August) onto the two normalized PCs across models ("Methods"), we find that the first EOF mode is linked to a cold-tongue-like SST anomalies in the central-eastern Pacific (180–80° W), stretching westwards from the coast of South America to the central Pacific (CP) along the equator (Fig. 1c). Although the pattern is not identical to the typical cold tongue bias which centers in the CP[23], it reflects most of the intermodel spread in the cold-tongue SST. This mode indicates that if a model has colder bias in the cold tongue region in the historical simulation, its projected future WNPSH tends to be weaker as shown in the EOF1 (Fig. 1a). The second EOF mode is associated with cold SST anomalies and more cloud cover over the eastern Pacific and Atlantic (Fig. 1d). These regions are usually covered with the well-known marine boundary layer clouds which play a pivotal role in cooling the Earth by reflecting solar radiation back to space. It is known that there are notably warm SST biases and a lack of low clouds over these regions in most climate models[18,20,21]. Thus, if a model has weaker positive SST biases in the eastern Pacific and Atlantic basins, the projected future WNPSH will tend to be stronger as shown in the EOF2 (Fig. 1b). Similar SST patterns associated with the PCs can also be found in the pre-industrial simulations which excludes any possible influences from external forcing (Supplementary Fig. 3), indicating a robust relationship between model SST biases in the current climate and spread in the WNPSH future projection.

**Constraining PC uncertainty using observational SST patterns.** The relationship between SST patterns and the leading modes of intermodel spread established above allows emergent constraints on the WNPSH projection, thanks to the relatively reliable SST observations. A well-established hierarchical statistical framework for emergent constraints[30] is used to derive a more robust projection than the conventional multi-model ensemble mean (MME) (see details in "Methods"). First, two indices are produced to measure the model's fidelity in simulating the observed climate in the historical period by projecting the mean-state SSTs onto the two SST modes associated with the intermodel leading PCs shown in Fig. 1c, d ("Methods"): T1 for the central-eastern Pacific cold tongue region (gray box in Fig. 1c) and T2 for the marine stratocumulus regions (gray box in Fig. 1d). These two indices are also calculated for the five observational SST datasets to compare with those for each model. T1 is larger than observation in more than 60% of the models (Fig. 2a), indicating that simulated SSTs in most models are too cold along the equatorial cold tongue but too warm outside the equator (Fig. 1c). The T2 index is smaller than observations or even negative in about 70% of the models (Fig. 2b), indicating a widespread warm bias in the marine stratocumulus regions.

A linear fitting across models between the current climate ($X$) and projected PCs ($Y$) is assumed: $Y = \bar{Y} + r(X - \bar{X})$ where $r$ is the regression coefficient and $\bar{X}$ and $\bar{Y}$ are the MME. In the hierarchical statistical framework under Gaussian assumptions, the $r$ is additionally influenced by the signal-noise ratio (SNR) in the observed current climate since the observations are involved in the constraints (see "Methods"). If the SNR $\gg 1$, its influence on $r$ can be neglected. For T1 and T2, the SNR is as high as $(17.4)^2$ and $(9.0)^2$, respectively. The high SNR benefits from the strong agreement in climatological SST patterns between the observational datasets (Supplementary Fig. 4). Therefore, the corrected

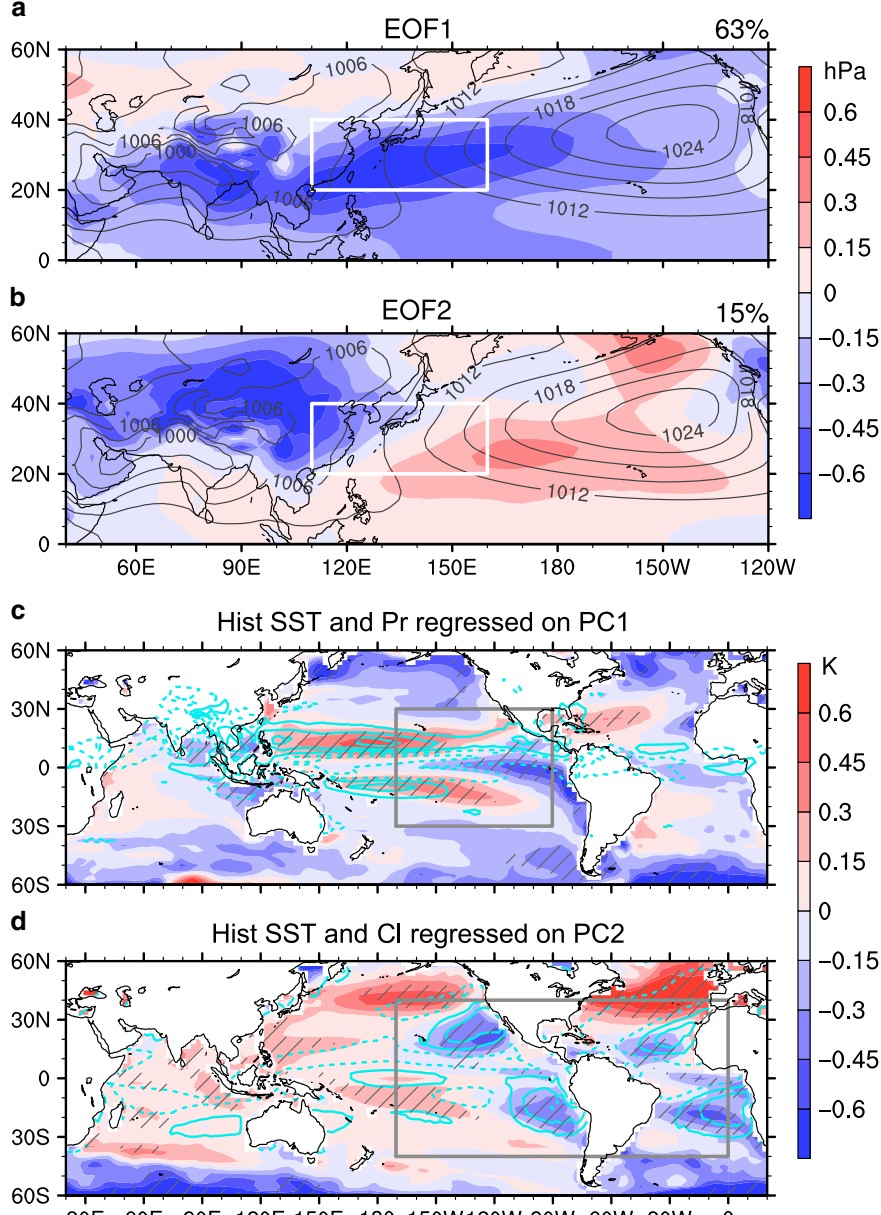

**Fig. 1 Projected leading uncertainty modes and related historical spread patterns. a**, **b** The two leading modes (EOF1 and EOF2) derived from intermodel empirical orthogonal function (EOF) analysis on projected changes of the western North Pacific Subtropical High (white box) under a high emission scenario (RCP8.5; see details in "Methods"), showing anomalous sea level pressure (SLP; shadings; hPa) by regressing onto the corresponding first and second normalized principal components (PC1 and PC2; Supplementary Fig. 2), overlaid by climatological SLP (contours; hPa). Value on the top-right corner is explained intermodel variance by corresponding mode. **c** Historical model spread patterns of sea surface temperature (SST; shading; K) and precipitation (Pr; contours drawn for ±0.4, ±1.2, and ±2.0 mm day$^{-1}$) associated with PC1, and **d** the patterns of SST (shading; K) and cloud fraction (Cl; contours drawn for ±2, ±6, and ±10%) associated with PC2. To exclude the influence of global-scale bias in SST simulation, the mean SST in 30°S–30°N is subtracted in each model before regressed onto the PCs. Gray boxes in (**c**) and (**d**) are used to define SST pattern indices to constrain the PCs ("Methods"). Hatched regions are statistically significant at the 5% level under Student t-test.

regression line (thin red line in Fig. 2) differs little from the original one (bold gray line).

Another important consequence of the hierarchical statistical framework yields the relative reduction in variance after constrained by observations (Eq. (6)). For PC1 and PC2, the reduced variances are about 59 and 52% (Fig. 3), respectively, mainly determined by the high correlation coefficients with T1 and T2 (Fig. 2) since the SNRs are large enough. Considering the explained variances by PC1 and PC2, 63 and 15%, respectively (Supplementary Fig. 2), the totally reduced variance by the emergent constraints is about 45% (59% × 63% + 52% × 15%).

Based on Eqs. (5) and (6), mean of the observed T1 and T2 yield corresponding optimal PCs, that is PC1 = −0.22 ± 0.64 and PC2 = 0.53 ± 0.70 (Fig. 3).

New (constrained) projections are produced by correcting the conventional MME using the EOF patterns and the optimal PCs (Fig. 4; "Methods"). Associated changes in precipitation and 850 hPa winds are also corrected ("Methods"). The constrained projections show a prominent enhancement of the WNPSH in the latter half of the 21st century (Fig. 4a), in contrast to the ambiguous results in the original MME (Supplementary Fig. 1). Consequently, with both the enhanced WNPSH and Asian Low,

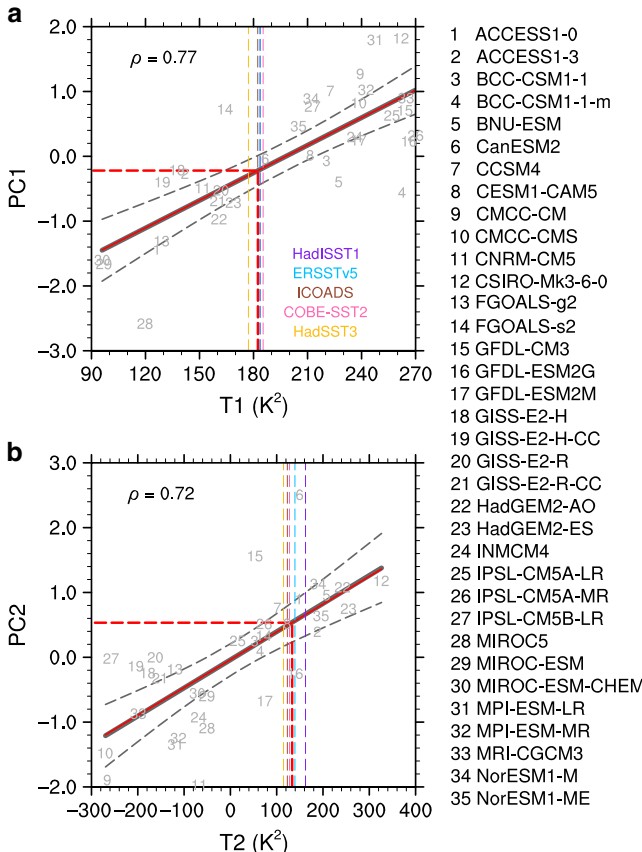

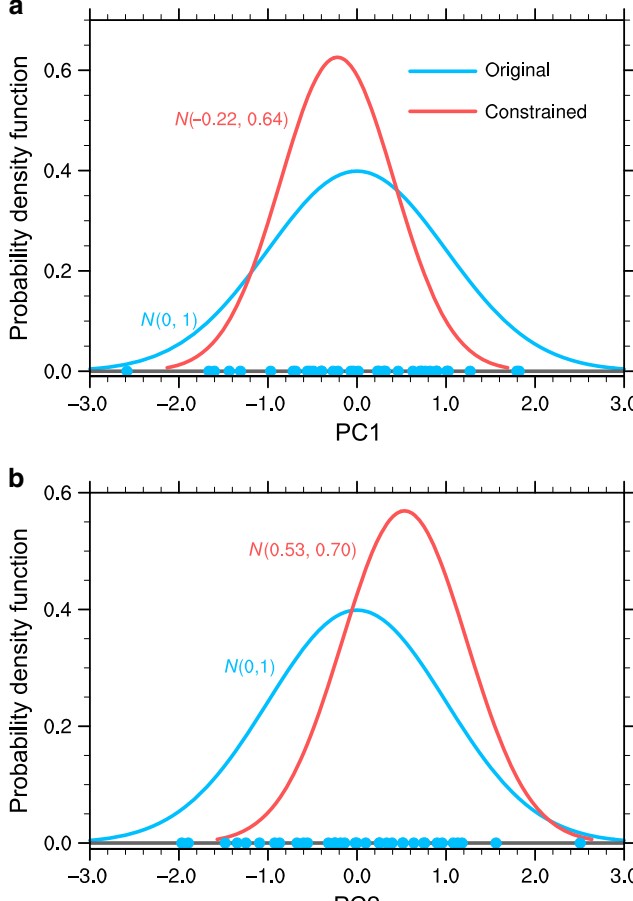

**Fig. 2 Relationship between spreads in projection and historical temperature pattern.** T1 in (**a**) and T2 in (**b**) ($K^2$) measure how the sea surface temperature (SST) patterns in gray boxes in Fig. 1c, d are simulated in a model's historical climate, respectively (see Eqs. (2) and (3)). T1 and T2 well explain the first and second principal components (PC1 and PC2), respectively, the two leading uncertainty modes of projected changes in the western North Pacific Subtropical High, with high correlation coefficients ($\rho$) statistically significant at the 1% level under Student $t$-test. Bold gray fitting line is obtained by the least square method while thin red line is an observational correction based on Eq. (5). Gray dashed curves denote the 95% confidence range of the linear regression. T1 and T2 indices from five observational SST datasets (HadISSTv1.1, ERSSTv5, ICOADS, COBE-SST2, and HadSST3; vertical thin dashed lines) are used to constrain the values of PCs. Mean of the five observational results yield the optimal constraint (red dashed line).

**Fig. 3 Probability density function of original and constrained principal components. a, b** Probability density functions of the first and second principal components (PC1 and PC2; Supplementary Fig. 2) are generated under Gaussian assumption. The PCs represent the leading intermodel uncertainty modes of the projected changes in the western North Pacific Subtropical High. The values in parentheses are mean and standard deviation of the Gaussian distribution. Dots denote the PC values of each model.

stronger monsoon southwesterly flows can bring more moisture into the Asian continent, leading to more rainfall in East Asia but less in Southeast Asia (Fig. 4a). Compared with uncorrected results, monsoon rain band in East Asia (called Mei-yu in China, Baiu in Japan, and Changma in Korea) is evidently enhanced (Fig. 4b). Under the control of stronger anticyclone over the WNP, convective activities and tropical storm genesis are expected to be suppressed. As a result, the overly projected rainfall in Southeast Asia and the WNP are also corrected (Fig. 4b). In the constrained results, about 28% more land rainfall is projected in East China (25–45°N, 105–120°E) and 17% more in the Korean peninsula and South Japan (30–40°N, 125–140°E), while about 36% less in Southeast Asia (10°S–20°N, 90–150°E) relative to the conventional MME. The less rainfall in Southeast Asia is contributed by both the two constrained EOFs whereas the more rainfall in East Asia is mainly contributed by the constrained EOF2 (Supplementary Fig. 5).

**Physical mechanisms backing the constraints.** To reveal the physical mechanisms that support the validity of the above emergent constraints on the WNPSH projection, we first examine the intermodel uncertainties in projected surface temperature changes by regressing the projected SST changes onto the PCs. The positive SST anomalies in the western Pacific (WP) are related to PC1 (Fig. 5a). SST biases in the cold tongue region were demonstrated to cause unrealistic warming in the WP[23,24]. The relationship between model uncertainty in historical simulation and future projection is explained by the following key physical processes. First, negative shortwave-SST (SW-SST) feedback related to convective clouds is an important mechanism to damp the local SST anomalies. A positive SST anomaly enhances convections over the equatorial Pacific, resulting in cloud increases that block incident SW and weaken the initial warming. In historical simulations across models, colder SST in the equatorial central-eastern Pacific induces less precipitation in the central-western Pacific (Fig. 6a). Less precipitation means fewer clouds which impairs the negative SW-SST feedback (Fig. 6c). Under GHG forcing in the future projection, less negative SW-SST feedback amplifies the warming in the equatorial WP (Fig. 6e). The formation of warm anomalies in the WP (Fig. 5a) additionally involves anomalous warm SST advection from the central

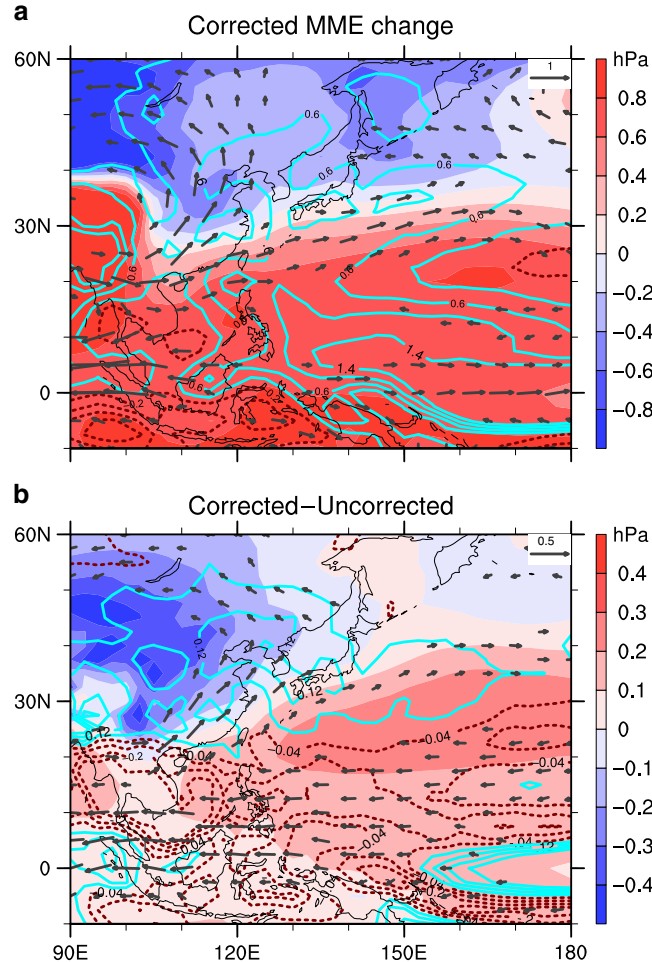

**Fig. 4 Corrected projections and differences from the uncorrected. a** Corrected sea level pressure projection (shadings; hPa) and related changes in 850 hPa wind (vectors drawn for larger than 0.2 m s⁻¹) and precipitation (contours; mm day⁻¹) over the western North Pacific region in the multi-model ensemble mean (MME) based on the best-estimated principal components (see details in "Methods"). **b** The difference between corrected and uncorrected results, which is reconstructed by the two constrained leading uncertainty modes (Supplementary Fig. 5). Corrected results show an enhanced western North Pacific Subtropical High, stronger monsoon circulation, and rainfall band in East Asia which are underestimated in the original MME.

to western Pacific by climatological westward equatorial current[24]. Second, the SST warming can intensify the convective heating over the western Pacific, triggering a pair of cyclonic anomalies in the low troposphere on both sides of the equator (Fig. 5c) through the Gill-type response[32], one of which is located over the WNP region (Fig. 6g). The baroclinic Rossby response can also be manifested by double warming centers and anticyclonic anomalies in the upper troposphere (Fig. 5e).

For the second leading mode, we find a global-scale warming pattern related to PC2 but with evident regional disparity, exhibiting large-scale land–sea thermal contrast and similar features to polar amplification (Fig. 5b and Supplementary Fig. 6a). This warming pattern coincides with the typical features associated with the global mean surface temperature (GMST) changes across models with a pattern correlation coefficient of 0.88 (Supplementary Fig. 6b). It implies that PC2 should be partly related to model spread in GMST projection or climate sensitivity. Intermodel spread in the shortwave low-cloud

feedback under GHG forcing dominates the uncertainty in modeled climate sensitivity[33]. Evident future changes in downward shortwave cloud radiation contributing to the spread in global mean warming are mostly in the regions where there are considerable shortwave reflective clouds (Supplementary Fig. 7). Given that 20% of the tropical oceans are covered by stratocumulus clouds[34] which reflects 30–60% of the incident shortwave radiation back to space[35], breakup of the marine stratocumulus clouds can result in huge global warming[36]. Colder SST beneath the marine stratocumulus in the historical simulation, accompanied with more clouds (Fig. 1d), can lead to larger positive shortwave cloud feedback under warming (Fig. 6b) because more cloud cover decreases in response to warming through feedbacks between low-cloud cover, low-cloud longwave radiative cooling, and relative humidity within the planetary boundary layer[22,25]. Hence, as expected, more positive shortwave cloud feedback in the marine stratocumulus regions induces larger GMST rise in the future projection across models (Fig. 6d). This mechanism explains the connection between the historical cold SST anomalies beneath the marine stratocumulus (Fig. 1d) and the global-scale warming related to PC2 (Fig. 5b).

Because of huge differences in heat capacity and evaporation between land and ocean, higher global mean warming induces larger thermal contrast between the Asian continent and subtropical North Pacific (Figs. 5b and 6f), reflected by enhanced sensible heating over the East Asian land (Fig. 5d). The amplified land–sea thermal contrast and the related diabatic heating leads to the intensification of both the Asian Low and WNPSH as shown in the EOF2 (Figs. 1b and 6h)[27,28]. In addition, subtropical westerly jets in the upper troposphere over East Asia and North Pacific are weakened due to the decreased meridional temperature gradient (Fig. 5f). The weakened subtropical jet stream and enhanced NPSH and WNPSH have been verified by the responses to enhanced land–sea thermal contrast under direct $CO_2$ forcing in a recent study using atmosphere-only models[29].

Key processes backing the emergent constraints on the two uncertainty modes of WNPSH projection are summarized in Fig. 7. About 45% of the CMIP5 intermodel variance is reduced by the emergent constraints. Through correcting the cold SST biases in the cold tongue and warm biases beneath the marine stratocumulus, we conclude that the WNPSH will robustly intensify with a westward extension in a future warming climate.

## Discussion

A westward shift of the WNPSH has been observed in the recent 4 decades[37]. Our results suggest that anthropogenic GHG forcing may contribute to this shift. An intensified WNPSH in future implies enhanced summer monsoon circulation with increased moisture transport farther into the inner Asian Continent, as well as increased risks of drought and flood to East Asia with seasonal march of the WNPSH but decreased typhoon landfalls.

Further tests carried out by using area-averaged SST instead of SST pattern or replacing SSTs from other parts of the world (Supplementary Fig. 8) have shown consistent results, demonstrating the robustness of the approach and conclusions presented above. The significant correlation between PC2 and historical North Atlantic and North Pacific SSTs across models (Fig. 1d and Supplementary Fig. 8b) is likely a reflection of model spread in simulating Atlantic Overturning Meridional Circulation (AMOC)[38] and the impact of AMOC response to GHG forcing on global mean warming[39,40]. It is well known that the AMOC has interhemispheric impact on temperature and precipitation[41,42]. How the historical SST biases in North Pacific and North Atlantic impact the WNPSH projection through AMOC simulation warrants further studies.

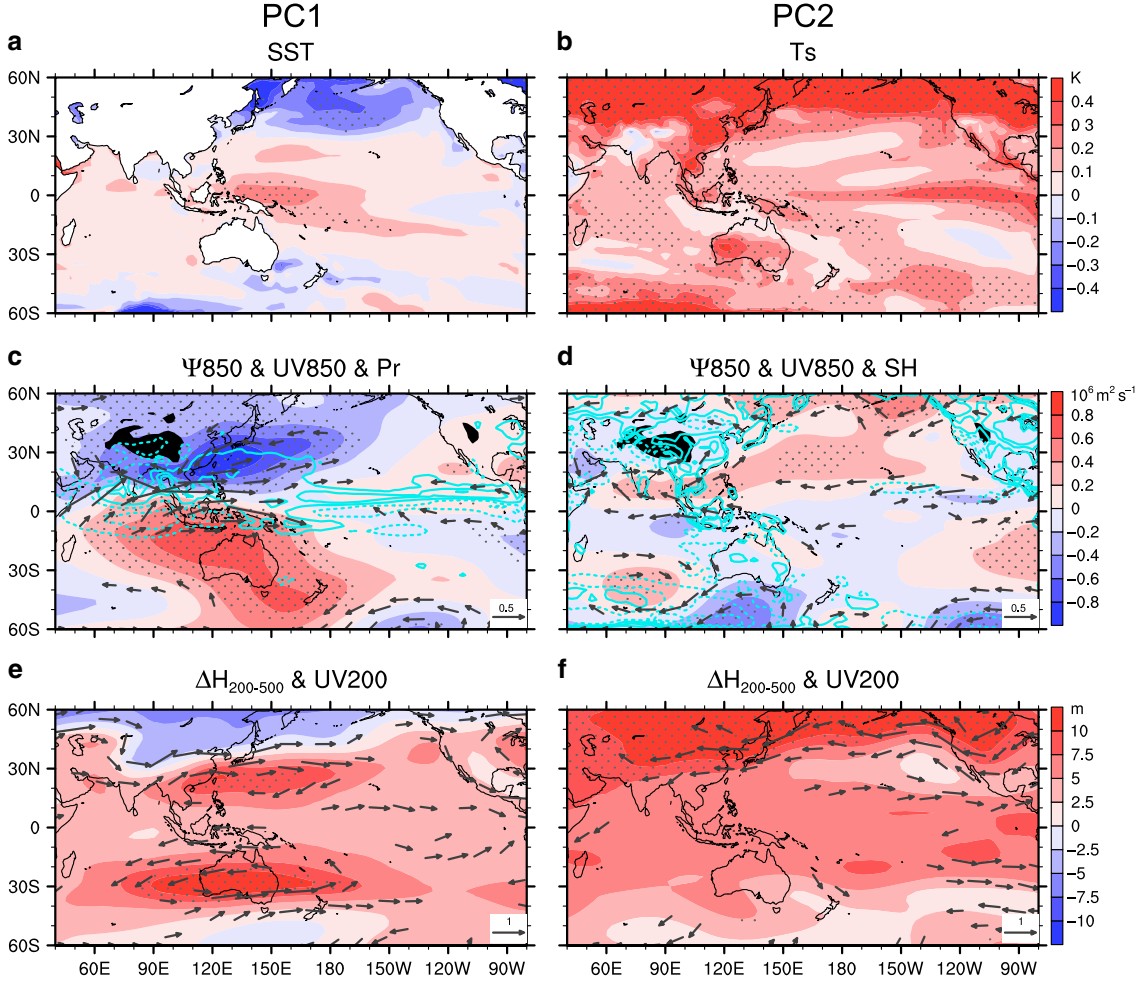

**Fig. 5 Physical mechanisms related to model uncertainties in projection. a**, **c**, **e** Intermodel spread in projected changes associated with the first principal component (PC1) and **b**, **d**, **f** those associated with the second principal component (PC2). **a**, **b** Sea surface temperature (SST; shadings; K) and surface temperature (Ts; shadings; K), respectively. **c**, **d** Stream function ($\psi$850; shadings; $10^6$ m$^2$ s$^{-1}$) and wind (UV850; vectors drawn for larger than 0.2 m s$^{-1}$) at 850 hPa, precipitation (Pr; contours in (**c**) drawn for ±0.2, ±0.6, and ±1.0 mm day$^{-1}$), and sensible heat flux (SH; contours in (**d**) drawn for ±0.6, ±1.8, and ±3.0 W m$^{-2}$). **e**, **f** Thickness ($\Delta$H$_{200-500}$; shadings; m) between 200 and 500 hPa and wind (UV200; vectors drawn for larger than 1 m s$^{-1}$) at 200 hPa. Dotted shadings are statistically significant at the 5% level under Student *t*-test. Uncertainty in the equatorial western Pacific warming in (**a**) leads to the first uncertainty mode through the Gill-type response in (**c**) and (**e**) by triggering convective heating over the western Pacific in (**c**). Uncertainty related to changes in land–sea thermal contrast in (**b**) and associated sensible heating in (**d**) are responsible for the second uncertainty mode. Weakened subtropical jet stream in (**f**) is also manifested as a result of enhanced land–sea thermal contrast.

After corrected by the emergent constraints, remaining uncertainty in the projected WNPSH can be partly attributed to other unconstrained modes and internal variability. The latter is climate noise which can hardly be constrained in long-term projection, accounting for 10–20% of the total uncertainty (Supplementary Fig. 9). Nevertheless, spread in the constrained PCs could be further narrowed if there were higher correlation coefficients between the quantities chosen to represent current climate and projected future changes. Other quantities independent of the SST patterns are suggested to be explored to improve the constraint efficacy.

## Methods

**Models and datasets**. Historical simulation and future projection under RCP8.5 scenario of 35 CMIP5 models are used (Supplementary Table 1). To represent future projection of the WNPSH, we focus on the changes of seasonal-mean sea level pressure during June, July, and August over the WNP region between 2050–2099 and 1956–2005. Because most CMIP5 models output only one realization for the RCP8.5 projection, the first realization (r1i1p1) of each model is used. The 50-year mean state is used to reduce the impact of internal variability as much as possible. Analysis on several single-model large-ensemble simulations

shows that contribution from internal variability to intermodel variance of the WNPSH projection is less than 20% (Supplementary Fig. 9). Sea level pressure (SLP), surface temperature (Ts, i.e., SST in the ocean), precipitation (Pr), cloud fraction (Cl), horizontal wind (U and V) at 850 and 200 hPa, geopotential height (Z) at 500 and 200 hPa, surface heat and radiative flux and radiation at the top of the atmosphere (TOA) are used in this study. To verify the robustness of the relationship between current model spread in SST patterns and future WNPSH projection by excluding any effect of external forcing, SST in pre-industrial control (piControl) simulation of each model is also used.

Five observational SST datasets are used to constrain the uncertainty modes of WNPSH projection. They are HadISST1[43], HadSST3[44], ERSSTv5[45], ICOADS[46], and COBE-SST2[47]. Same time period of 1956–2005 as the model baseline is used to calculate mean state in the observation. All data are remapped onto a 2.5° grid by bilinear interpolation.

**Intermodel empirical orthogonal function analysis**. The leading modes of intermodel uncertainty in the WNPSH projection in the domain (20–40°N, 110–160°E) is analyzed by the typical EOF method, applied to model-spatial dimension:

$$\Delta SLP'(m,s) \cong \sum_{1}^{n}\left(PC_{i,m}\times EOF_{i,s}\right),\qquad(1)$$

in which $\Delta$ denotes projected changes, $m$ the model number, $s$ the spatial grid, and $n$ the mode number. Prime means the deviation from the multi-model ensemble

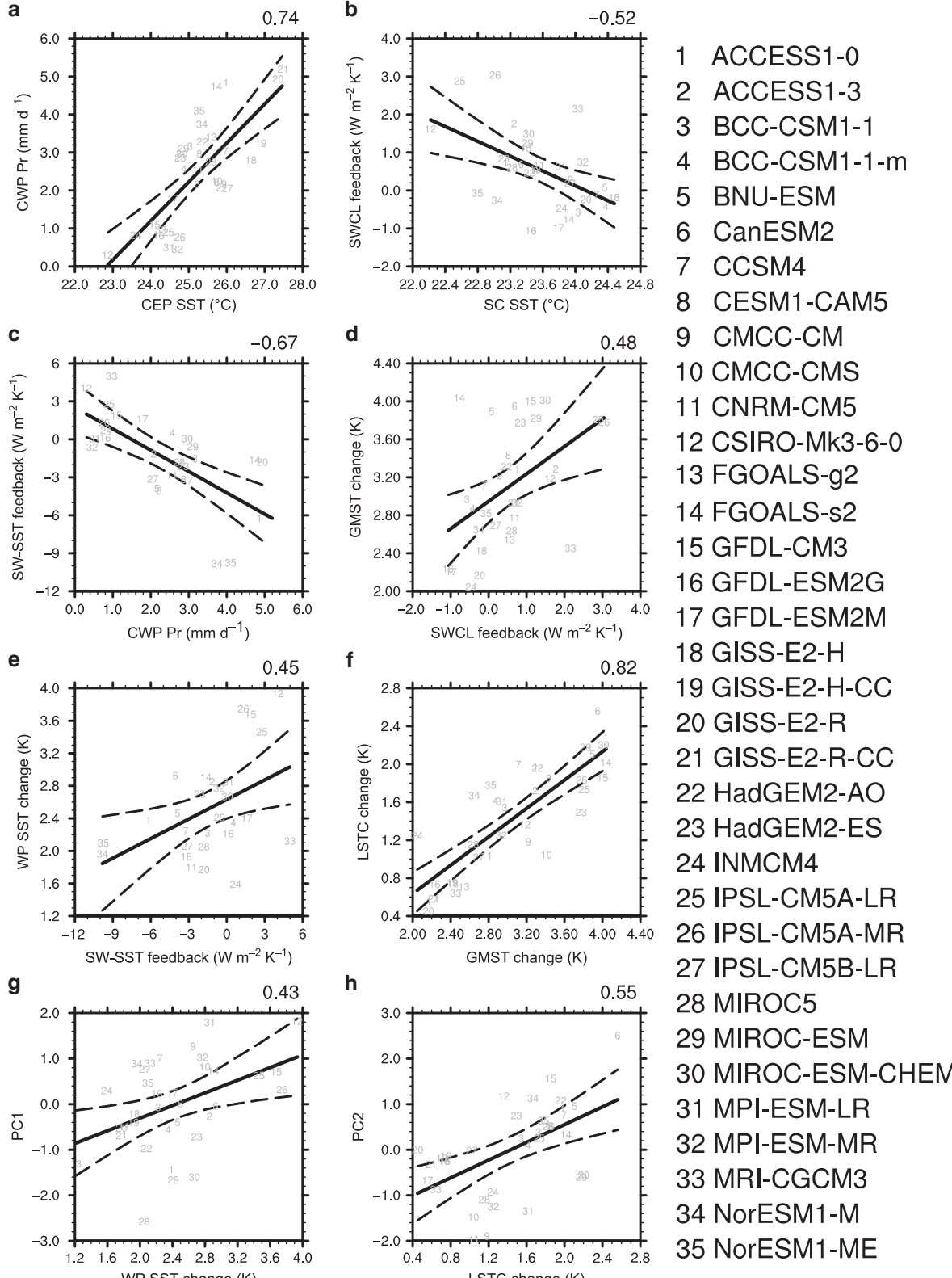

**Fig. 6 Pipelining physical links between historical and projected spreads. a**, **c**, **e**, **g** Intermodel relationship among the central-eastern Pacific (CEP) sea surface temperature (SST), central-western Pacific (CWP) precipitation, cloud shortwave-SST (SW-SST) feedback, western Pacific (WP) SST change, and the first principal component (PC1). **b**, **d**, **h** Intermodel relationship among SST beneath the marine stratocumulus (SC), shortwave cloud (SWCL) feedback, global mean surface air temperature (GMST) change, and the second principal component (PC2). The indices above are defined in "Methods". Solid fitting line is obtained by the least square method. Dashed curves denote the 95% confidence range of the linear regression. Value on the top-right corner of each subplot is correlation coefficient. All the correlation coefficients are statistically significant at the 5% level under Student $t$-test.

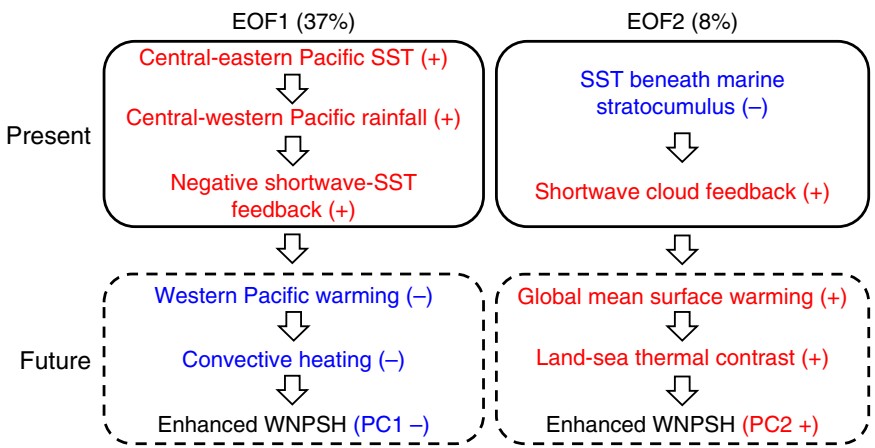

**Fig. 7 Physical processes backing emergent constraints on uncertain modes.** Solid boxes denote processes in the historical period and dashed boxes for future projection. Red: a process is larger/stronger in observation than historical simulation in multi-model ensemble mean (MME) or it should be larger/stronger in future changes than original projection in MME. Blue: opposite to the red. Percentages on the top are reduced variances after constrained by observational sea surface temperature (SST) for the two leading intermodel empirical orthogonal function (EOF) modes (EOF1 and EOF2) of projected changes in the western North Pacific Subtropical High (WNPSH). The constrained results for both the two modes favor an enhanced western North Pacific Subtropical High in future, represented by a negative value of the first principal component (PC1) and a positive value of the second principal component (PC2).

mean. Here, PCs are normalized. The eigenvalue in each mode is merged in corresponding EOF.

**Definition of SST pattern indices.** Two SST pattern indices, T1 and T2, are defined to represent current climate, quantifying the SST distributions in the central-eastern Pacific cold-tongue region (30°S–30°N, 90–170°W; Fig. 1c) and in the stratocumulus cloud regions (40°S–40°N, 90–360°W; Fig. 1d) to constrain the PC1 and PC2 of intermodel spread in WNPSH projection, respectively. Large domains rather than local area are used to more clearly reflect the SST patterns. T1 and T2 are calculated by projecting historical mean state SST in boreal summer in each model ($SST_{hist}$) onto the intermodel anomalous SST ($SST'_{PC}$; Fig. 1c, d) in the focused regions using the scalar product as

$$T1 = SST_{hist} \cdot SST'_{PC1} \quad (30°S\text{–}30°N, \ 90\text{–}170°W); \tag{2}$$

$$T2 = SST_{hist} \cdot SST'_{PC2} \quad (40°S\text{–}40°N, \ 90\text{–}360°W). \tag{3}$$

For calculating observational T1 and T2, $SST_{hist}$ is derived from the five observational SST datasets.

To reduce the influence of model spread in global tropical mean SST and amplify the signal of large-scale SST pattern, the historical mean SST between 30°S and 30°N is subtracted in each model and observation before calculating the two indices.

**Hierarchical statistical framework for emergent constraint.** Reference [30] proposed a hierarchical framework for emergent constraints in strict statistical theory. Here, the framework is described in a more practical way that is directly related to this study. More details and applications of the method can be seen in ref. [30].

In the framework, first we should establish a dependence between future climate change $Y$ and current climate $X$. The uncertainty in $Y$ is the target to be constrained. A simple straight-line approximation between $Y$ and $X$ can be obtained from climate model ensembles, that is

$$Y = \bar{Y} + r(X - \bar{X}), \tag{4}$$

where $r = \frac{\sigma_Y}{\sigma_X}\rho$ the regression coefficient, $\rho$ the correlation coefficient between $Y$ and $X$, and $\sigma_Y$ and $\sigma_X$ the standard deviation of $Y$ and $X$ across models, respectively; $\bar{X}$ and $\bar{Y}$ are multi-model ensemble mean. In this study, the $Y$ are the first two normalized leading PCs (i.e., $\bar{Y} = 0$ and $\sigma_Y = 1$) and $X$ the corresponding SST pattern indices defined in Eqs. (2) and (3).

Since the observational current climate $X_O$ is used to constrain the $Y$, the uncertainty in the observations should be considered. With an additive-noise model under Gaussian assumptions that relates the observations to current climate[30], the signal-noise ratio (SNR) in the observed current climate is derived to correct the scaling factor $r$ by multiplying $\frac{1}{1+SNR^{-1}}$, where $SNR = \sigma_X^2/\sigma_O^2$ in which $\sigma_X^2$ is estimated by the variance across models and $\sigma_O^2$ the variance across different observational datasets. If SNR is large enough (i.e., $SNR \gg 1$), the effect of correction can be neglected. Hence, combining Eq. (4) and the SNR correction,

constrained expectation, and variance of future climate change $Y_C$ can be expressed as

$$\overline{Y_C} = \bar{Y} + \frac{r}{1 + SNR^{-1}}\left(\overline{X_O} - \bar{X}\right); \tag{5}$$

$$\sigma_{Y_c}^2 = \left(1 - \frac{\rho^2}{1 + SNR^{-1}}\right)\sigma_Y^2. \tag{6}$$

Finally, relative reduction in variance $\left(1 - \frac{\sigma_{Y_c}^2}{\sigma_Y^2}\right)$ derived from the hierarchical statistical framework is $\frac{\rho^2}{1+SNR^{-1}}$. Thus, in this study, totally reduced model variance (TRV) by constraining the PC1 and PC2 can be expressed as weighting on the corresponding explained variances PCV1 (63%) and PCV2 (15%) in percentage:

$$TRV = \frac{\rho^2_{PC1,T1}}{1 + SNR_{T1}^{-1}}PCV1 + \frac{\rho^2_{PC2,T2}}{1 + SNR_{T2}^{-1}}PCV2. \tag{7}$$

If $SNR \gg 1$, the reduced variance is only determined by $\rho$, or the impact of SNR cannot be neglected.

**Correction on multi-model mean projection.** Optimal PC1 and PC2 are estimated by the emergent constraint using mean of the observed T1 and T2 (Fig. 2). Then WNPSH projection is corrected based on EOF reconstruction following Eq. (1):

$$\Delta SLP = \Delta\overline{SLP} + \Delta SLP' \approx \Delta\overline{SLP} + \sum_1^n (PC_{i,O} \times EOF_{i,s}); \tag{8}$$

in which subscript "O" denotes optimal PCs constrained by observational SST; bar denotes the multi-model ensemble mean. Here, mode number $n$ is 2. The precipitation and 850 hPa wind fields can be corrected in a similar way to Eq. (8), but the EOF terms are replaced by regression coefficients related to corresponding PCs.

**Defining indices of key physical processes.** To illustrate possible physical mechanisms supporting the emergent constraints on WNPSH projection, several indices are defined based on previous literature or the results in this study.

To understand the relationship between intermodel spread in the historical SST in the cold-tongue region and PC1, the following indices are defined by averaging in corresponding domains in boreal summer.

(1) Equatorial central-eastern Pacific (CEP) SST in historical simulation: 2°S–2°N, 180–80°W.
(2) Equatorial central-western Pacific (CWP) precipitation in historical simulation: 2°S–2°N, 150°E–120°W.
(3) Local cloud shortwave-SST feedback in the equatorial CWP: $\Delta R^{\downarrow}_{SWCL}/\Delta SST$; 2°S–2°N, 150°E–120°W. $\Delta$ means projected changes between RCP8.5 and historical simulations. $R^{\downarrow}_{SWCL}$ is surface downward shortwave cloud (SWCL) radiation, calculated by the all-sky minus the clear-sky outputs.
(4) Equatorial western Pacific (WP) SST change: $\Delta SST$; 2°S–2°N, 130–170°E.

To understand the relationship between intermodel spread in the historical SST beneath the marine stratocumulus region and PC2, the following indices are defined by averaging in corresponding domains in boreal summer.

(5) SST beneath the marine stratocumulus (SC) region in historical simulation: 30°S–30°N, 150°W–360°, excluding the domain between 10°S and 10°N.

(6) Shortwave cloud (SWCL) feedback: $\Delta F^{\downarrow}_{SWCL}/\Delta GMST$; 30°S–30°N, 140°W–360°. $F^{\downarrow}_{SWCL}$ is net downward shortwave cloud radiation at TOA, calculated by the all-sky minus the clear-sky outputs.

(7) Land–sea thermal contrast (LSTC) change: projected change of surface temperature difference between the Asian continent (20–60°N, 60–120°E) and the subtropical western North Pacific (20–40°N, 130°E–150°W).

## Data availability

CMIP5 model data are from the Earth System Grid Federation [https://esgf-node.llnl.gov/projects/cmip5/]. Observational SST data HadISST1 and HadSST3 are from Met Office Hadley Centre [https://www.metoffice.gov.uk/hadobs/]. ERSSTv5, ICOADS, and COBE-SST2 are provided by the NOAA/OAR/ESRL PSD, Boulder, CO, USA [https://www.esrl.noaa.gov/psd/data/gridded/tables/sst.html].

## Code availability

The data in this study is analyzed with NCAR Command Language (NCL [http://www.ncl.ucar.edu/]). All relevant codes used in this work are available, upon request, from the corresponding author (X.C.).

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

## Acknowledgements

The study is jointly supported by the Strategic Priority Research Program of the Chinese Academy of Sciences (Grant No. XDA20060102), the National Natural Science Foundation of China (Grant No. 41605057), and the International Partnership Program of Chinese Academy of Sciences (Grant No. 134111KYSB20160031). P.W. was supported by the UK–China Research & Innovation Partnership Fund through the Met Office Climate Science for Service Partnership (CSSP) China as part of the Newton Fund.

## Author contributions

X.C. designed the research, performed the analysis, and drafted the manuscript. P.W., T.Z., M.W., and Z.G. gave comments and helped revise the manuscript. All of the co-authors contributed to scientific interpretations.

## Competing interests

The authors declare no competing interests.
