## [Peer Review File · Nature Communications]

Reviewers' comments first round:

Reviewer #1 (Remarks to the Author):

The paper is well written with sound methodology and contains useful scientific information on the North Pacific Subtropical High. The paper is recommended for publication with minor suggestions.

Suggestions:

A recent study (Preethi et al 2017a) has shown that the North Pacific Subtropical High has shifted westwards about 5-7 degrees of longitudes during recent decades. A further study (Preethi et al 2017b) showed that this High depicts a zonal oscillatory behavior with east-west-east shifts in the future projections. The authors could consider citing these papers

References:

Preethi B, M Mujumdar, RH Kripalani, A Prabhu, R Krishnan 2017a: Recent trends and teleconnections among South and East Asian summer monsoons in a warming environment. *Climate Dynamics*, 48, 2489-2505.

Preethi B, M Mujumdar, A Prabhu, RH Kripalani 2017b: Variability and teleconnections of South and East Asian summer monsoons in present and future projections of CMIP5 climate models. *Asia-Pacific J of Atmos Sc*, 53(2), 303-325

Reviewer #2 (Remarks to the Author):

Review of manuscript #NCOMMS-19-3009875 entitled "Emergent constraints on future projections of the western North Pacific Subtropical High".

Chen et al. use an ensemble of global climate models to identify a reasonably strong intermodel relationship between historical sea surface temperatures and projected changes to the western North Pacific Subtropical High (WNPSH). Observations of sea surface temperature across two regions are then used in conjunction with this relationship in an attempt to narrow intermodel spread in the leading modes of the WNPSH projection. The authors use this constraint to conclude that the WNPSH will strengthen under future warming. This anticipated intensification is shown to have major implications for East Asian climate. The research is novel and would be of interest to the climate community. However, the manuscript is fairly difficult to read. I have pointed out several instances of this below. I recommend asking a native English speaker to assist with proofreading during revision.

Major comments:

1) Other reviewers will be better informed to comment on the underlying large-scale dynamics, but the authors do a good job of highlighting the existence of a relationship between historical sea surface temperature patterns and projected changes to the WNPSH. However, there are additional steps that would help to solidify the proposed emergent constraint. The statistical methodology is generally adequate to the current level of development for emergent constraints, although the emergent projection can be improved through the incorporation of a methodology such as that of Bowman et al. (2018, <https://agupubs.onlinelibrary.wiley.com/doi/full/10.1029/2018GL080082>). Given that the authors have a series of observational datasets (and thus, can determine observational uncertainty), a more robust projection of the optimal PC1 and PC2 can be derived using this methodology.

2) Similarly, it is important to know what factors (model processes/characteristics or parameterizations) are driving the intermodel spread in simulated present-day SST patterns across the T1 and T2 regions. Without this knowledge, it is difficult for modelling centers to attempt to fix

the bias and thus, improve future projections. While a full investigation of these factors is outside the scope of this study, a preliminary analysis would be highly beneficial. Given that space is limited, some simple scatterplots related to this (between simulated historical SST patterns and relevant factors/processes) could be shown as Supplemental Figure(s).

3) Role of internal variability: The authors do not state whether they are using the first ensemble member for each CMIP5 model or not? Assuming that this is the case, a note should be added to the Methods. Furthermore, it brings up the question of how much sensitive the results are to internal variability. I am interested in seeing how sensitive a model's placement on Figure 2 is to the choice of ensemble member. This simple analysis could be performed for a single model with several CMIP5 realizations and added as a supplemental figure if the results warrant inclusion.

Specific comments:

L40: Change "despite of intensive" to "despite a number of intensive"

L47: Remove "state-of-the-art" here, this is already stated in the paragraph above.

L50: Add "the" before "surface warming pattern"

L56 and L64: Change "projection" to "projections"

L57: Change "75% uncertainty" to "75% of the uncertainty"

L66: Change "consistence" to "consistency". I am also confused what is meant by model consistency in this context, does it have to do with the sign of future change? Please clarify and rephrase in the text.

L70: Remove "totally".

FigS1: Add a panel showing the historical SLP over the same region to give the reader more context.

L83: change "present-day simulation" to "historical simulation" throughout.

Fig 1: State what the interval between contours represents, is it 1 mm/day and 1% change in cloud fraction? Also, why is the mean SST being taken over different areas (30-30 and 60-60) before being regressed onto the principal components?

L88: Use "It has been shown" or "It is known".

L89: Change "too few low-cloud covers" to "a lack of low clouds"

L92: Add some information about the pre-industrial simulation(s) to the Methods. Is this the PI-control scenario from each CMIP5 model?

L99-100: I thought the stratocumulus cloud regions had a cold bias (as stated on L85-86)? Please clarify.

Fig 2: I am somewhat surprised to see such strong agreement between the observational datasets. I recommend creating a supplemental figure showing maps of their SST patterns over the historical period for T1 and T2.

Calculation of T1 and T2 indices: I would like to see more detail regarding how these indices are derived. In the Fig 2 caption the authors note that "Negative T1 denotes a reduction of cold tongue bias...", but the language used is confusing. Is the index centered on the model ensemble mean value? If so, "anomaly" is more appropriate than "bias", which is often reserved when comparing models to observations.

Fig 1ab,3,4: It is difficult to see the land boundary on many of these maps. Perhaps switch the SLP contour color to grey and the land boundary to black for figs 1a and b?

L125: I appreciate this discussion of the physical mechanisms behind the constraint, but would change the subtitle to "Physical mechanisms backing the constraint".

L129: "The positive SST warming" should say "The positive SST anomalies" or "The SST warming"

L148: Change "can result larger decrease" to "can result in larger decreases"

L149: change "respond" to "response"

L175: "Reduce the CMIP5 intermodel spread by 75%" 75% of what? Please clarify and rephrase.

L179: Fix "father"

L209-211: Expand on why the global-scale bias in SSTs must be subtracted.

L212-213: There are more robust approaches to estimating the optimal PC1 and PC2 than just using the mean of observed T1 and T2 (see main comment #2).

Reviewer #3 (Remarks to the Author):

COMMENT SHEET

for

PAPER NUMBER: NCOMMS-19-3009875

JOURNAL: Nature Communications

TITLE: Emergent constraints on future projections of the western North Pacific Subtropical High

The manuscript by Chen et al. present an interesting analysis and interpretation on future projection of western North Pacific Subtropical High (WNPSH). The study describes two inter-model modes that can explain uncertainties of future changes in WNPSH simulated by CMIP5 climate models. They found that the first leading mode is linked to the models' cold tongue biases in the tropical Pacific and the second mode is linked to the warm biases associated with marine stratocumulus. They corrected future projection of WNPSH using observational constraints estimated through the principal component and the related sea surface temperature pattern. The study further suggested the physical mechanisms behind the two leading inter-model modes. The paper is well focused, and the study objectives are clear. The finding is new and could be potentially important to understand future changes in WNPSH and east Asian monsoon. However, although the authors try to present a compelling case, the amount of information provided and the failure to support their interpretation weakens their case. The present study is mostly based on descriptive analysis, lacks important details, and must be strengthened before it can be accepted for publication. The major and minor issues are detailed below.

1.The critical point is the appropriateness of the analysis used in this study. The key analysis in this paper is to find optimal principal components (PCs) from the inter-model relationship between leading PCs and sea surface temperature (SST) pattern indexes as shown in Fig. 2. However, there seems to be circular reasoning in this analysis given that the SST pattern index (i.e., T1 and T2) is based on PCs. By the definition of T1, PC1 and T1 are expected to be correlated to each other, and thus I'm wondering how the result can be justified to constraint future model prediction of WNPSH.

2.The pattern of cold tongue bias stretching to the coast of Peru as shown in Fig 1c is not similar to that of the typical cold tongue bias of climate models. How would the result change if T1 is defined by equatorial SST mean in different models? And for the T2, how would be the result if T2 is defined by area-averaging the warm SST bias in the subtropics?

3.Another important issue is that physical mechanisms suggested in the study are not complete,

rather the arguments are largely based on previous works and speculation. Particularly, the linkage between cold tongue SST bias in the historical run and future warming in the western Pacific is not clearly shown. The linkage between warm SST bias (or low cloud bias) and enhance land-sea thermal contrast also needs to be strengthened. Isn't there possibility that models showing stronger Arctic amplification can cause stronger low cloud and warm SST biases? The causal relationship is still not clear in the mechanism related to PC2. Overall, the mechanism section seems to overly speculative given the tenuous arguments presented.

4.The methodology used in this study has not been explained in sufficient detail. The authors explained that T1 and T2 are the projections of historical mean SST pattern onto the SST pattern associated with EOF1 and EOF2, respectively. Does the projection refer to the regression coefficient between them? Is the historical mean SST from the observations subtracted from each modeled SST to define T1 and T2? In line 210, the historical mean SST refers to one value averaged in the specified area?

5.Other minor comments are as follows.

-Line 72: EOF2 doesn't look like enhanced WNPSH.

-Line 90: fewer \diamond weaker

-Line 102: What is the degree of freedom to calculate p-value? In the multi-model set used in the study, the models rooted in the same family are used, thus it seems the degree of freedom should not be simply N-2.

-Future change in circulation pattern related to EOF2 (Fig. 4) do not really indicate the enhanced WNPSH. Again, is the second inter-model mode related to WNPSH?

-Authors' argument on heat waves, drought, and flood is unrelated to the topic and probably better to be avoided because there is not any analysis on these topics.

-Last paragraph seems to be redundant and duplicated.

The original comments of reviewers are in *italic*. The revised parts are marked in **red in the tracked-change version of manuscript.**

Reviewers' comments:

Reviewer #1 (Remarks to the Author):

The paper is well written with sound methodology and contains useful scientific information on the North Pacific Subtropical High. The paper is recommended for publication with minor suggestions.

Suggestions:

A recent study (Preethi et al 2017a) has shown that the North Pacific Subtropical High has shifted westwards about 5-7 degrees of longitudes during recent decades. A further study (Preethi et al 2017b) showed that this High depicts an zonal oscillatory behavior with east-west-east shifts in the future projections. The authors could consider citing these papers

References:

Preethi B, M Mujumdar, RH Kripalani, A Prabhu, R Krishnan 2017a: Recent trends and tele-connections among South and East Asian summer monsoons in a warming environment. Climate Dynamics, 48, 2489-2505.

Preethi B, M Mujumdar, A Prabhu, RH Kripalani 2017b: Variability and tele-connections of South and East Asian summer monsoons in present and future projections of CMIP5 climate models. Asia-Pacific J of Atmos Sc, 53(2), 303-325

Response: The studies you mentioned extend the scope of influence of the WNPSH. The observed westward shift of WNPSH may partly include external forcing, consistent with the result in this study. We cited them in the revised manuscript.

“Zonal oscillatory behaviour of the WNPSH under the green-house gas (GHG) forcing exacerbates the uncertainty in the future projection.” (L48-49)

“Our results indicate that anthropogenic forcing signal may partly contribute to the observed westward shift in the recent four decades.” (L215-216)

Reviewer #2 (Remarks to the Author):

Review of manuscript #NCOMMS-19-3009875 entitled “Emergent constraints on future projections of the western North Pacific Subtropical High”.

Chen et al. use an ensemble of global climate models to identify a reasonably strong intermodel relationship between historical sea surface temperatures and projected changes to the western North Pacific Subtropical High (WNPSH). Observations of sea surface temperature across two regions are then used in conjunction with this relationship in an attempt to narrow intermodel spread in the leading modes of the WNPSH projection. The authors use this constraint to conclude that the WNPSH will strengthen under future warming. This anticipated intensification is shown to have major implications for East Asian climate. The research is novel and would be of interest to the climate community. However, the manuscript is fairly difficult to read. I have pointed out several instances of this below. I recommend asking a native English speaker to assist with proofreading during revision.

Response: Thank you for your comments. We have responded to your concerns one by one below and carefully made a word editing.

Major comments:

1) Other reviewers will be better informed to comment on the underlying large-scale dynamics, but the authors do a good job of highlighting the existence of a relationship between historical sea surface temperature patterns and projected changes to the WNPSH. However, there are additional steps that would help to solidify the proposed emergent constraint. The statistical methodology is generally adequate to the current level of development for emergent constraints, although the emergent projection can be improved through the incorporation of a methodology such as that of Bowman et al. (2018, <https://agupubs.onlinelibrary.wiley.com/doi/full/10.1029/2018GL080082>).

Given that the authors have a series of observational datasets (and thus, can determine observational uncertainty), a more robust projection of the optimal PC1 and PC2 can be derived using this methodology.

Response: Thank you for your suggestion. The results are updated by using the hierarchical statistical framework according to **Ref. 28** you recommended. The section “Constraining PC uncertainty using observational SST patterns” is substantially enriched (**L109-138**). The core of this approach is described in the Methods (**L264-295**). The new method does not change the main results and conclusions in the previous manuscript except several corrections. The total reduced variance can be more accurately estimated in the hierarchical statistical framework (~45%), which was overestimated in the previous version. We added a new figure (**Fig. 3**) to show the constrained PCs derived from the framework. Because the signal-to-noise ratio in observational dataset are large enough, $(7.8)^2$ and $(9.0)^2$ for the T1 and T2 indices, respectively, the constrained result is mostly determined by the correlation coefficient between the SST pattern indices and PCs.

Bowman, K. W., Cressie, N., Qu, X., & Hall, A. A hierarchical statistical framework for emergent constraints: Application to snow-albedo feedback. *Geophysical Research Letters* **45**, 13050–13059 (2018).

2) Similarly, it is important to know what factors (model processes/characteristics or parameterizations) are driving the intermodel spread in simulated present-day SST patterns across the T1 and T2 regions. Without this knowledge, it is difficult for modelling centers to attempt to fix the bias and thus, improve future projections. While a full investigation of these factors is outside the scope of this study, a preliminary analysis would be highly beneficial. Given that space is limited, some simple scatterplots related to this (between simulated historical SST patterns and relevant factors/processes) could be shown as Supplemental Figure(s).

Response: Thank you for your suggestion. To reveal the processes that dominate the

model spread in historical SST, surface energy fluxes are analyzed based on the balance between emitted longwave (LWE) from surface and other energy sources/sinks, including cloud radiative effect, oceanic dynamics, sensible and latent heat, that is

$$R_{LWE} = R_{LWCL} + R_{SWCL} + R_{LWCS} + R_{SWCS} + SH + LH + OD.$$

The left-hand term is the longwave emitted from surface, reflecting the SST spread itself (Figs. R1a and R2a below). The emitted longwave is balanced by radiative fluxes (cloud longwave: LWCL; cloud shortwave: SWCL; clear-sky longwave: LWCS; clear-sky shortwave: SWCS), heat fluxes (sensible heat: SH; latent heat: LH) and oceanic dynamic effect (OD). The OD term is calculated as a residual. Positive direction is upward for the left-hand term and downward for the right-hand terms. Hence, the model spreads in historical LWE (i.e. SST) can be explained by different contributions of the right-hand processes. If a process is positively correlated to the LWE across models, it can be regarded as a possible driver or positive feedback to amplify SST, otherwise it is a negative feedback to damp the SST.

For the SST in the central-eastern Pacific, based on the results, oceanic dynamics, clear-sky longwave and shortwave, and sensible heat flux could significantly contribute to the model spread (Fig. R1). For the SST underneath the marine stratocumulus, important processes are oceanic dynamics, cloud shortwave and clear-sky longwave flux (Fig. R2). The ocean dynamics include the Bjerknes feedback related to the strength of upwelling in the eastern Pacific (Li et al. 2016), and oceanic eddy mixing and diffusion between upper layer and deep ocean in the stratocumulus regions (Richter 2015; Zuidema et al. 2016). Nevertheless, the clear-sky longwave process which has the closest intermodel relationship with the surface thermal cooling (Figs. R1e and R2e), suggesting an important role of water vapor feedback in the local SST spread. In contrast, latent heat flux always acts as a negative feedback (Figs. R1h and R2h).

While the scatterplots help to understand the factors that are driving the intermodel spread, a full investigation calls for further study. Since the motivation of this study is to constrain the projection by using the current state-of-the-art models instead of the model improvement, we did not add Figs.R1, R2 in the SI and the corresponding the

discussion in the text, as you noted that the manuscript *is already fairly difficult to read*. However, we would be happy to add them to the manuscript in liaison with you and the editors if you strongly encourage us.

Li, G., Xie, S.-P., Du, Y. & Luo, Y. Effects of excessive equatorial cold tongue bias on the projections of tropical Pacific climate change. Part I: the warming pattern in CMIP5 multi-model ensemble. *Climate Dynamics* **47**, 3817–3831 (2016).

Richter, I. Climate model biases in the eastern tropical oceans: causes, impacts and ways forward. *WIREs Climate Change* **6**, 345–358 (2015).

Zuidema, P. *et al.* Challenges and prospects for reducing coupled climate model SST biases in the eastern tropical Atlantic and Pacific Oceans: The U.S. CLIVAR Eastern Tropical Oceans Synthesis Working Group. *Bulletin of the American Meteorological Society* **97**, 2305–2328 (2016).

Fig. R1 Intermodel relationship between SST and surface energy fluxes in the equatorial central-eastern Pacific (2°S – 2°N , 180 – 80°W) in historical simulation.

(a) The model spread in SST can be well represented by emitted longwave from surface which can be explained by (b-h) other energy fluxes based on Equation (9) in Methods. Positive direction is upward for the emitted longwave from surface, but downward for other flux processes. The solid fitting line is obtained by the least square method.

Dashed curves denote 95% confidence range of the linear regression. Values on top of each subplot denote regression coefficient with $\pm 1\sigma$ and correlation coefficient. A star marks correlation coefficient exceeding the 5% significance level.

Fig. R2 The same as Fig. R1 but for the marine stratocumulus region (30°S – 30°N , 150°W – 360° , excluding the domain between 10°S – 10°N).

3) *Role of internal variability: The authors do not state whether they are using the first ensemble member for each CMIP5 model or not? Assuming that this is the case, a note should be added to the Methods. Furthermore, it brings up the question of how much sensitive the results are to internal variability. I am interested in seeing how sensitive a model's placement on Figure 2 is to the choice of ensemble member. This simple analysis could be performed for a single model with several CMIP5 realizations and added as a supplemental figure if the results warrant inclusion.*

Response: Only the first realization (r1i1p1) of each CMIP5 model was used because most models only output this one realization for the RCP8.5 projection. Based on your suggestion, here we use three large ensemble simulations of CanESM2 (50 members), CESM (40 members) and MPI-ESM-MR (100 members) to show some preliminary results. As shown in **Fig. R3** below (**added in as Supplementary Fig. 7**), the spread in WNPSH projection across CMIP5 models is two more times larger than that from internal variability. If measured by variance ratio ($\sigma_{internal}^2/\sigma_{intermodel}^2$), the contributions from internal variability is less than 20%. Considering that the emergent constraints reduce about 45% uncertainty of PCs in this study, the remaining uncertainty may partly be contributed by internal variability. We added a comment on this in discussions (**L209-211**) and a brief explanation in the Methods (**L226-229**).

Fig. R3 Spread in WNPSH projection (SLP averaged in 20–40°N, 110–160°E) in 35

CMIP5 models and in large ensemble simulations of three single models, CanESM2 (50 members), CESM (40 members) and MPI-ESM-MR (100 members). The spread ($\pm 1\sigma$) in each large ensemble simulation is caused by internal variability which is less than half of the spread caused by model structure. If measured by variance ratio ($\sigma_{internal}^2/\sigma_{intermodel}^2$), the contributions from internal variability is less than 20%.

Specific comments:

1) L40: Change “*despite of intensive*” to “*despite a number of intensive*”

Response: Done (L44). We also changed the expression during word editing.

2) L47: Remove “*state-of-the-art*” here, *this is already stated in the paragraph above.*

Response: Done.

3) L50: Add “*the*” before “*surface warming pattern*”

Response: Done (L56).

4) L56 and L64: Change “*projection*” to “*projections*”

Response: Done (L58, L62).

5) L57: Change “*75% uncertainty*” to “*75% of the uncertainty*”

Response: Done (L66). We changed the expression during word editing.

6) L66: Change “*consistence*” to “*consistency*”. *I am also confused what is meant by model consistency in this context, does it have to do with the sign of future change? Please clarify and rephrase in the text.*

Response: The expression is rephrased to explicitly show the meaning as follows, “where less than 70% of the models agree on the sign of change (Supplementary Fig. 1a) and the signal-to-noise ratio is below 0.5 (Supplementary Fig. 1b)” (L74-76).

7) L70: Remove “*totally*”.

Response: Done.

8) FigS1: Add a panel showing the historical SLP over the same region to give the reader more context.

Response: The historical SLP distribution has been plotted, with the cyan contours overlaying on the shadings (**Supplementary Fig. 1a**).

9) L83: change “*present-day simulation*” to “*historical simulation*” throughout.

Response: Done (**L93, L105**).

10) Fig 1: State what the interval between contours represents, is it 1 mm/day and 1% change in cloud fraction? Also, why is the mean SST being taken over different areas (30-30 and 60-60) before being regressed onto the principal components?

Response: The contour values are added in the figure caption (**L474-475**). Our idea of removing a global-scale mean SST from the original historical climatology is to highlight the role of SST pattern rather than absolute values. Here we find the method does not affect the results for PC1. Hence, in the revised manuscript, we use the original SST to derive the SST pattern related to the PC1. For PC2, model spread in tropical mean SST impacts the robustness of intermodel relationship between PC2 and historical SST, although the SST pattern is very similar no matter a global-scale mean SST is removed or not (**Fig. R4**). The correlation coefficient between the averaged SST in the marine stratocumulus region (box in Fig. R2) and PC2 can be as high as -0.55 when the mean SST in 30°S-30°N is removed, or it is only -0.33. A statement is added in Methods to explain why a global-scale mean SST is removed for deriving the SST pattern related to the PC2 (**L261-263**).

Fig. R4 Historical SST pattern related to the PC2 of uncertainty in WNPSH projection. (a) Mean SST in 30°S-30°N is subtracted from original SST in each model before regressing onto the PC2 while (b) uses the original SST.

11) L88: Use “It has been shown” or “It is known”.

Response: Done (L98).

12) L89: Change “too few low-cloud covers” to “a lack of low clouds”

Response: Done (L99).

13) L92: Add some information about the pre-industrial simulation(s) to the Methods.

Is this the PI-control scenario from each CMIP5 model?

Response: Done (L234-237).

14) L99-100: *I thought the stratocumulus cloud regions had a cold bias (as stated on L85-86)? Please clarify.*

Response: The cold SST pattern in Fig. 1d is the intermodel spread pattern related to PC2. It only indicates the relationship of intermodel spread between current climate and future projection. If the opposite PC2 (-PC2) is used, a warm SST pattern in the stratocumulus cloud regions will be seen. In fact, most models simulate a warmer SST beneath the stratocumulus than the observations. In **L98-102**, we clearly point out the fact and how it could imply the observational constraint. Because the observed SST beneath the stratocumulus cloud regions is colder than the modelled, the constrained PC2 has a positive value (**Fig. 7**).

15) Fig 2: *I am somewhat surprised to see such strong agreement between the observational datasets. I recommend creating a supplemental figure showing maps of their SST patterns over the historical period for T1 and T2.*

Response: We added **Supplementary Fig. 4** to show the strong agreement in climatological SST pattern in observational datasets. The strong agreement ensures a high signal-to-noise in the current climate (**L126-127**).

16) *Calculation of T1 and T2 indices: I would like to see more detail regarding how these indices are derived. In the Fig 2 caption the authors note that “Negative T1 denotes a reduction of cold tongue bias...”, but the language used is confusing. Is the index centered on the model ensemble mean value? If so, “anomaly” is more appropriate than “bias”, which is often reserved when comparing models to observations.*

Response: The confusing sentence has been deleted. Detailed calculation of T1 and T2 are described in Methods (**L250-260**). In the revised manuscript, the indices are calculated based on Equations (2) and (3) in Methods without any postprocessing, which impacts little on the results.

17) Fig 1a,b,3,4: It is difficult to see the land boundary on many of these maps. Perhaps switch the SLP contour color to grey and the land boundary to black for figs 1a and b?

Response: Revised (**Figs. 1, 3 and 4**).

18) L125: I appreciate this discussion of the physical mechanisms behind the constraint, but would change the subtitle to “Physical mechanisms backing the constraint”.

Response: Corrected (**L158**).

19) L129: “The positive SST warming” should say “The positive SST anomalies” or “The SST warming”

Response: Corrected (**L161**).

20) L148: Change “can result larger decrease” to “can result in larger decreases”

Response: The sentence is reorganized (**L186-190**).

21) L149: change “respond” to “response”

Response: Corrected (**L189**).

22) L175: “Reduce the CMIP5 intermodel spread by 75%” 75% of what? Please clarify and rephrase.

Response: Corrected. Based on the hierarchical statistical framework, reduced variance is overestimated in the original study. In the revised manuscript, it is corrected to 45% (**L209**).

23) L179: Fix “father”

Response: Corrected (**L218**).

24) L209-211: Expand on why the global-scale bias in SSTs must be subtracted.

Response: As explained in a response above, our idea of removing a global-scale mean SST from the original historical climatology is to highlight the role of SST pattern rather than absolute values. Here we find the method does not affect the results for PC1. In Methods we now add the explanation why a global-scale mean SST is removed for deriving the SST pattern related to the PC2. More details can be found in the response (item #10) related to **Fig. R4 (L261-263)**.

25) L212-213: *There are more robust approaches to estimating the optimal PC1 and PC2 than just using the mean of observed T1 and T2 (see main comment #2).*

Response: Thanks for your suggestion. We now employ your recommended approaches to make a more robust constraint (**L109-138**). The core of this approach is summarized in the Methods (**L264-295**). The new method has small effect on the constrained results because the signal-to-noise ratio is very high for the climatological SST pattern in current climate, whereas it can give a more accurate reduced variance.

Reviewer #3 (Remarks to the Author):

COMMENT SHEET

for

PAPER NUMBER: NCOMMS-19-3009875

JOURNAL: Nature Communications

TITLE: Emergent constraints on future projections of the western North Pacific Subtropical High

The manuscript by Chen et al. present an interesting analysis and interpretation on future projection of western North Pacific Subtropical High (WNPSH). The study describes two inter-model modes that can explain uncertainties of future changes in WNPSH simulated by CMIP5 climate models. They found that the first leading mode is linked to the models' cold tongue biases in the tropical Pacific and the second mode is linked to the warm biases associated with marine stratocumulus. They corrected future projection of WNPSH using observational constraints estimated through the principal component and the related sea surface temperature pattern. The study further suggested the physical mechanisms behind the two leading inter-model modes.

The paper is well focused, and the study objectives are clear. The finding is new and could be potentially important to understand future changes in WNPSH and east Asian monsoon. However, although the authors try to present a compelling case, the amount of information provided and the failure to support their interpretation weakens their case. The present study is mostly based on descriptive analysis, lacks important details, and must be strengthened before it can be accepted for publication. The major and minor issues are detailed below.

Response: Thank you for your comments. We have responded to your concerns one by one below.

1. The critical point is the appropriateness of the analysis used in this study. The key

analysis in this paper is to find optimal principal components (PCs) from the inter-model relationship between leading PCs and sea surface temperature (SST) pattern indexes as shown in Fig. 2. However, there seems to be circular reasoning in this analysis given that the SST pattern index (i.e., T1 and T2) is based on PCs. By the definition of T1, PC1 and T1 are expected to be correlated to each other, and thus I'm wondering how the result can be justified to constraint future model prediction of WNPSH.

Response: The method of emergent constraint requires establishing the relation between current climate and future projection (please see the description of emergent constraint in Methods which has been substantially updated based on the 1st comment of Reviewer #2; **L264-295**). We hope to clarify that while the high correlations ensure a robust constraint to reduce the model uncertainty, the chosen regions should be physically based. Inspection on Figs. 1c-d finds high correlation coefficient in many locations, only the central-eastern Pacific and marine stratocumulus regions are investigated in our study since they are physically based. To clearly present the logic, we first introduce how to look for the relationship to constrain uncertainty, and then we provide the physical explanations in the manuscript.

2. The pattern of cold tongue bias stretching to the coast of Peru as shown in Fig 1c is not similar to that of the typical cold tongue bias of climate models. How would the result change if T1 is defined by equatorial SST mean in different models? And for the T2, how would be the result if T2 is defined by area-averaging the warm SST bias in the subtropics?

Response: Yes, the historical SST pattern related to PC1 is not identical to the typical cold tongue bias which centers in the equatorial central Pacific. Since the significant anomalies are indeed located in the central-eastern Pacific cold tongue region (180–80°W) and like the climatological SST pattern (**Supplementary Fig. 4**), we now call it cold-tongue-like SST pattern in the revised manuscript (**L26**).

As you suggested, we define a SST index in the equatorial central-eastern Pacific

(CEP; 2°S–2°N, 180–80°W) to represent the cold-tongue-like pattern. Intermodel correlation coefficient between the CEP SST and a typical cold tongue index (2°S–2°N, 180–140°W; Woelfle et al. 2018) is as high as 0.90. A brief explanation is added (**L90-92**). For the SST pattern related to PC2 (Fig. 1d), we define a subtropical SST index in the eastern North Pacific and North Atlantic (ENP_NA; 35–45°N, 180–10°W).

The scattering plots of the two indices, CEP and ENP_NA against PC1 and PC2, respectively, are shown in **Fig. R5** below. The newly constrained results (PC1 = -0.19 ± 0.95 ; PC2 = 0.42 ± 0.86) are close to those constrained by T1 and T2 (PC1 = -0.23 ± 0.64 ; PC2 = 0.53 ± 0.70), which verifies that the emergent constraints in this study are robust. The CEP SST, as an important source to the PC1 uncertainty (**Fig. 6a**), can be used to constrain PC1 directly (**Fig. R5a**), as well as the ENP_NA SST to constrain PC2 (**Fig. R5b**). But the constraint is less effective than that constrained by the SST pattern because of relatively low correlation coefficient. Why the SST patterns used in the emergent constraints are more efficient than the absolute SST is still an open question. We would be happy to add the discussion above to the manuscript in liaison with you and the editors if you strongly encourage us.

Woelfle, M. D., Yu, S., Bretherton, C. S., & Pritchard, M. S. Sensitivity of coupled tropical pacific model biases to convective parameterization in CESM1. *Journal of Advances in Modeling Earth Systems* **10**, 126–144 (2018).

Fig. R5 Intermodel relationship between leading PCs and absolute sea surface temperature in historical simulation. Equatorial central-eastern Pacific (CEP) SST (°C) is an index defined in 2°S–2°N, 180–80°W (Methods) and the eastern North Pacific and North Atlantic (ENP_NA) SST index (°C) is defined in 35–45°N, 180–

10°W. Correlation coefficients (ρ) exceed the 5% significance level. The bold grey fitting line is obtained by the least square method while the thin red line is a correction based on Equation (5). Black dashed curves denote 95% confidence range of the linear regression. The indices from five observational SST datasets (purple line: HadISSTv1.1; blue line: ERSSTv5; green line: ICOADS; pink line: COBE-SST2; orange line: HadSST3) are used to constrain the values of PCs. Mean of the five observational results gives the optimal constraint (red dashed line): PC1 = -0.19 ± 0.95 ; PC2 = 0.42 ± 0.86 .

3. Another important issue is that physical mechanisms suggested in the study are not complete, rather the arguments are largely based on previous works and speculation. Particularly, the linkage between cold tongue SST bias in the historical run and future warming in the western Pacific is not clearly shown. The linkage between warm SST bias (or low cloud bias) and enhance land-sea thermal contrast also needs to be strengthened. Isn't there possibility that models showing stronger Arctic amplification can cause stronger low cloud and warm SST biases? The causal relationship is still not clear in the mechanism related to PC2. Overall, the mechanism section seems to overly speculative given the tenuous arguments presented.

Response: Thank you for your comments. The logic of emergent constraint is to constrain future projection based on relationship between spread in historical simulation and future projection. Hence, when illustrating the physical mechanism backing the constraints, we start from historical simulation and then look for reasonable processes step by step, ultimately to fill the gap between the uncertainty in current climate and future change. We enriched the section of physical mechanisms to make it more complete (L161-205). Based on previous studies, the key physical processes connecting the historical SST spreads with the PCs are clearly shown in a new scattering plot, **Fig. R6** (also added as **Fig. 6** in the revised manuscript). Corresponding descriptions and explanations are added in the text.

For the first mode, the importance of negative shortwave-SST (SW-SST) feedback

is highlighted (**L165-172**) which establishes the link between cold tongue SST bias in the historical run and future warming in the western Pacific (**Left column in Fig. R6**). For the second mode, the associated Arctic amplification is the well-known spatial feature of global warming. There are plenty of studies addressing why the Arctic region warms more than other places under the GHG forcing. Considering the similar warming pattern related to PC2 with that related to GMST change (**Supplementary Fig. 6**), it is reasonable to come up with an idea to explain the spread in PC2 by global mean warming. We clearly show the physical chains from the current cold SST with more clouds in the marine stratocumulus region to future high global warming and large land-sea thermal contrast (**L178-205; Right Column in Fig. R6**).

Based on our current knowledge, it is difficult to understand how stronger Arctic amplification, which usually means an exceptional warming in the Arctic region and surrounding continents, can cause stronger low cloud and warm SST biases. Nevertheless, as shown in **Fig. R5b**, the North Atlantic and North Pacific SSTs (Fig. 1d) can be also used to constrain PC2. But the underlying mechanism remains unclear. A possible explanation involves the Atlantic Overturning Meridional Circulation (AMOC). Warmer SST in the North Atlantic and North Pacific in the current climate is related to stronger AMOC (Zhang and Zhao 2015), implying that most CMIP5 models may underestimate the AMOC strength (**Fig. R5b**). A stronger AMOC, however, tends to weaken more under CO₂ forcing (He et al. 2017), thus leading to weaker ocean heat uptake efficiency and higher global surface warming (Chen and Tung 2018), which can explain the PC2 uncertainty (Figs. 6f and 6h). However, a weaker AMOC could also lead to a higher global warming because it weakens less from CO₂ forcing which results in smaller reduction in polarward heat transport (He et al. 2017). Hence, the roles of SST biases in the extratropical oceans and underlying mechanisms in the projection uncertainty are still under debate and deserve further study. Since there are controversial conclusions on the effect of AMOC in uncertainty of global warming, we are very cautious in adding it to the manuscript. However, we would be happy to add the discussion above to the manuscript in liaison with you and the editors if you strongly

encourage us.

Zhang, L. & Zhao, C. Processes and mechanisms for the model SST biases in the North Atlantic and North Pacific: A link with the Atlantic meridional overturning circulation, *Journal of Advances in Modeling Earth Systems* **7**, 739–758 (2015).

He, J., Winton, M., Vecchi, G., Jia, L. & Rugenstein, M. Transient climate sensitivity depends on base climate ocean circulation. *Journal of Climate* **30**, 1493–1504 (2017).

Chen, X. Y. & Tung, K.-K. Global surface warming enhanced by weak Atlantic overturning circulation. *Nature* **559**, 387–391 (2018).

Fig. R6 Pipelining physical processes to explain the relationship between simulated historical SST spread and the two leading uncertainty modes of WNPSH projection. The left column is for EOF1/PC1 showing intermodel relationship among the central-eastern Pacific (CEP) SST, central Pacific (CP) precipitation, cloud shortwave-SST (SW-SST) feedback, western Pacific (WP) SST change and PC1, and the right column for EOF2/PC2 including pipelining relationship among SST beneath

the marine stratocumulus (SC), cloud shortwave (SWCL) feedback, global mean surface temperature (GMST) change, land-sea thermal contrast (LSTC) change and PC2. Definitions of the indices above are described in Methods. The solid fitting line is obtained by the least square method. Dashed curves denote 95% confidence range of the linear regression. The value on top right corner of each subplot is correlation coefficient. All the correlation coefficients exceed the 1% significance level.

4. The methodology used in this study has not been explained in sufficient detail. The authors explained that T1 and T2 are the projections of historical mean SST pattern onto the SST pattern associated with EOF1 and EOF2, respectively. Does the projection refer to the regression coefficient between them? Is the historical mean SST from the observations subtracted from each modeled SST to define T1 and T2? In line 210, the historical mean SST refers to one value averaged in the specified area?

Response: The definition of SST pattern indices has been revised and are now more clearly described in Methods (L250-260). Before calculating the T2 index, one value of SST averaged in 30°S–30°N is subtracted from the original field to reduce the influence of systematic bias in global tropical SST and amplify the signal of large-scale SST pattern (L261-263). Please also see our responses to Reviewer#2 (Specific comments, Item#10 of Reveiwer#2).

5. Other minor comments are as follows.

-Line 72: EOF2 doesn't look like enhanced WNPSH.

Response: In Fig. 1b, we can see the main body of the subtropical high is over the ocean. Positive SLP anomalies are dominant over the western North Pacific. In contrast, the SLP over the Asian continent decrease. Hence, the EOF2 indicate that both the Asia Low and WNPSH are enhanced (L81-83). Because the white box includes part of the continent, the enhanced Asian Low and WNPSH largely offset in the selected domain. That is why a larger scale pattern is shown in Fig. 1b.

-Line 90: fewer → weaker

Response: Corrected (L100).

-Line 102: What is the degree of freedom to calculate p-value? In the multi-model set used in the study, the models rooted in the same family are used, thus it seems the degree of freedom should not be simply $N-2$.

Response: Strictly speaking, the degree of freedom should be less than $N-2$. However, it is difficult to estimate the accurate degree of freedom. The correlation coefficients in Fig. 2 are still significant ($p < 0.001$) even if the degree of freedom is reduced to half of the original (about 15). Nevertheless, we delete the p-value in the text.

-Future change in circulation pattern related to EOF2 (Fig. 4) do not really indicate the enhanced WNPSH. Again, is the second inter-model mode related to WNPSH?

Response: The increased SLP over the western North Pacific represent the enhanced WNPSH (Fig. 5d). The anticyclone anomalies over the WNP region and enhanced southerlies over the coast of East Asia are clear signals of enhanced WNPSH (Supplementary Fig. 5b) which has been traditionally regarded as the key component of the East Asian summer monsoon (e.g. Wang et al. 2008). As explained above, simultaneous intensification of the Asian Low and WNPSH resulting from enhanced land-sea thermal contrast makes the dipolar anomaly (see Fig. 1c in Shaw and Voigt, 2015 shown as Fig. R7 below).

Wang, B., Z. Wu, Jianping. Li, Jian Liu, C.-P. Chang, Y. Ding, and G.-X. Wu, 2008: How to Measure the Strength of the East Asian Summer Monsoon? *J. Climate*, 21, 4449-4463.

Shaw, T. A. & Voigt, A. Tug of war on summertime circulation between radiative forcing and sea surface warming. *Nature Geoscience* 8, 560–566 (2015).

Fig. R7 This plot is a copy of Fig. 1c from Shaw and Voigt, 2015. It shows the response of 925 hPa stationary eddy streamfunction (colour shading) in AMIP4xCO₂ experiment. Black contour lines are climatology. AMIP4xCO₂ is a sensitivity run using only AGCM with prescribed present-day SST as boundary condition under 4xCO₂ forcing. Hence, the experiment creates a very warm land and strong land-sea thermal contrast.

-Authors' argument on heat waves, drought, and flood is unrelated to the topic and probably better to be avoided because there is not any analysis on these topics.

Response: Deleted.

-Last paragraph seems to be redundant and duplicated.

Response: Deleted.

REVIEWER COMMENTS second round

Reviewer #2 (Remarks to the Author):

2nd Review of manuscript #NCOMMS-19-3009875A entitled "Emergent constraints on future projections of the western North Pacific Subtropical High"

The authors have done a good job of addressing my primary concerns. The resulting manuscript is much improved and fit for publication. I especially appreciate the added robustness associated with the hierarchical statistical framework, the new figures, and the more complete investigation of physical mechanisms driving the emergent relationship. I only have some brief technical corrections at this time.

L32: Change "those in the simple" to "suggested by the"

L44: change "remain" to "remaining"

L53: Remove "there are" since this is only two of many persistent biases in GCMs. "Two well-known chronic biases in current climate models are a cold tongue ..."

L66: Add a citation when mentioning emergent constraints here (e.g., Hall et al 2019; <https://www.nature.com/articles/s41558-019-0436-6>)

L206-209: Breakup into two sentences after "systematic model biases". A second sentence along these lines would be appropriate: "This constraint suggests a reduction in the CMIP5 intermodel variance by about 45% and is supported by sound physical mechanisms."

Reviewer #3 (Remarks to the Author):

PAPER NUMBER: NCOMMS-19-3009875A

JOURNAL: Nature Communications

TITLE: Emergent constraints on future projections of the western North Pacific Subtropical High

The authors have effectively responded to my previous comments and the manuscript has been much improved. However, some parts of the manuscript are still difficult to read, and it lacks some important details. I think it will likely be publishable after further revisions.

1. Please add more detailed information on the methodology used in the present study. Some of the detailed comments are as follows.

- Line 87: Please specify the season of SST patterns (I guess it's summer, right?)

- Lines 111-114: I think the methodology to define T1 and T2 should be understandable for readers in sufficient detail without reading the method section. It would be much better to add more detailed explanation here.

- Line 161-162: The method how to derive Fig. 5a is still unclear. Please add more detailed explanation. Does the "related" mean the projection of future SST changes (i.e., rcp85 run - historical run) onto PC1 or PC2?

- Lines 257: I guess observational SST dataset is not included to define T1 or T2. If I'm right, the sentence is misleading. If I'm wrong, a more detailed description on how to define T1 and T1 indices is still needed.

2. The physical interpretation from Fig. 6a to Fig 6e needs to be more clear. There seems to be a logical jump in Fig 6e. How does the historical SW-SST feedback relate to future WP SST warming? In particular, the SST warming center associated with PC1 appears in 120E-180E (Fig. 5a), but the region for SW-SST feedback is located further east. Is there any physical explanation for that?

3. The linkage between SW-cloud feedback and GMST also needs to be strengthened. Particularly,

given that future surface warming pattern shown in Fig. 5b is quite different from cloud fraction pattern shown in Fig. 1d, a more detailed mechanism how the local marine stratocumulus can change global mean SST should be added. I think the mechanism section is important to justify the emergent constraint method used in the present study, thus author needs to elaborate on it.

A few more comments are as follows:

Line 173: precipitation heating \diamond convective heating

Line 175: Gill response \diamond Gill-type response

Line 255: Does the 'absolute SST' means 'area-averaged SST'?

Line 261-263: I think the author should choose either subtracting the mean tropical SST or just using original SST when defining T1 and T2. The low correlation coefficient between PC2 and T2 when using original SST cannot justify the methodology and it gives an impression that the method used in this study is arbitrary.

Reviewer #2 (Remarks to the Author):

2nd Review of manuscript #NCOMMS-19-3009875A entitled “Emergent constraints on future projections of the western North Pacific Subtropical High”

The authors have done a good job of addressing my primary concerns. The resulting manuscript is much improved and fit for publication. I especially appreciate the added robustness associated with the hierarchical statistical framework, the new figures, and the more complete investigation of physical mechanisms driving the emergent relationship. I only have some brief technical corrections at this time.

1. L32: Change “those in the simple” to “suggested by the”

Response: Done (L32).

2. L44: change “remain” to “remaining”

Response: Done (L44).

3. L53: Remove “there are” since this is only two of many persistent biases in GCMs.

“Two well-known chronic biases in current climate models are a cold tongue ...”

Response: Done (L53).

4. L66: Add a citation when mentioning emergent constraints here (e.g., Hall et al 2019; <https://www.nature.com/articles/s41558-019-0436-6>)

Response: Done (L65, L449).

5. L206-209: Breakup into two sentences after “systematic model biases”. A second sentence along these lines would be appropriate: “This constraint suggests a reduction in the CMIP5 intermodel variance by about 45% and is supported by sound physical mechanisms.”

Response: Done (L220-221).

Reviewer #3 (Remarks to the Author):

PAPER NUMBER: NCOMMS-19-3009875A

JOURNAL: Nature Communications

TITLE: Emergent constraints on future projections of the western North Pacific Subtropical High

The authors have effectively responded to my previous comments and the manuscript has been much improved. However, some parts of the manuscript are still difficult to read, and it lacks some important details. I think it will likely be publishable after further revisions.

1. Please add more detailed information on the methodology used in the present study.

Some of the detailed comments are as follows.

- Line 87: Please specify the season of SST patterns (I guess it's summer, right?)

Response: Yes. Done. (L86)

- Lines 111-114: I think the methodology to define T1 and T2 should be understandable for readers in sufficient detail without reading the method section. It would be much better to add more detailed explanation here.

Response: We added a more detailed explanation (L111-117).

- Line 161-162: The method how to derive Fig. 5a is still unclear. Please add more detailed explanation. Does the "related" mean the projection of future SST changes (i.e., rcp85 run – historical run) onto PC1 or PC2?

Response: You are right. We revised the words and made it clearer (L164).

- Lines 257: I guess observational SST dataset is not included to define T1 or T2. If I'm right, the sentence is misleading. If I'm wrong, a more detailed description on how to define T1 and T1 indices is still needed.

Response: Observational dataset should be used to calculate corresponding T1 and T2 to constrain the model uncertainty (dashed colored lines in Fig. 2). Both the observational and models' indices are calculated based on Equations (2) and (3) in Methods. For example, the observational T1 is calculated based on Equation (2) in which SST_{hist} is the historical mean state SST pattern derived from observational dataset. We revised the words to make it clearer (L278-285).

2. *The physical interpretation from Fig. 6a to Fig 6e needs to be more clear. There seems to be a logical jump in Fig 6e. How does the historical SW-SST feedback relate to future WP SST warming? In particular, the SST warming center associated with PCI appears in 120E-180E (Fig. 5a), but the region for SW-SST feedback is located further east. Is there any physical explanation for that?*

Response: Thank you for your comment. To make it more consistent, we define a SW-SST feedback in the equatorial central-western Pacific (2°S – 2°N , 150°E – 120°W ; L337-338) which covers the warming center in the western Pacific in Fig. 5a. It impacts little on the results (Figs. 6c and 6e). To establish a connection between historical model spread in the eastern Pacific SST and future spread in the western Pacific warming, the central Pacific is a key region which conveys the signal from the eastern Pacific to the western Pacific. Previous study by Ying et al. (2019) has shown that the initial warming caused by a weak negative SW-SST feedback in the central Pacific could be advected westwards by mean equatorial current, leading to the warming center in the western Pacific. We added an explanation for the signal shift from the central to western Pacific (L176-178).

Ying, J., Huang, P., Lian, T. & Tan, H. Understanding the effect of an excessive cold tongue bias on projecting the tropical Pacific SST warming pattern in CMIP5 models. *Climate Dynamics* **52**, 1805–1818 (2019).

3. *The linkage between SW-cloud feedback and GMST also needs to be strengthened. Particularly, given that future surface warming pattern shown in Fig. 5b is quite*

different from cloud fraction pattern shown in Fig. 1d, a more detailed mechanism how the local marine stratocumulus can change global mean SST should be added. I think the mechanism section is important to justify the emergent constraint method used in the present study, thus author needs to elaborate on it.

Response: The cloud fraction pattern shown in Fig. 1d cannot directly explain the future surface warming pattern shown in Fig. 5b, because the latter results from land-sea distribution and local air-sea and air-land interactions. But the corresponding shortwave cloud feedback in the marine stratocumulus regions can indeed explain the GMST change through affecting the radiative energy balance at the top of the atmosphere in the global-mean perspective, since stratocumulus clouds cover 20% of the low-latitude oceans and determine a large part of energy into the earth system. We added a new plot as Supplementary Fig. 7 (Fig. R1 below) to show that evident future changes in downward shortwave cloud radiation contributing to the spread in global mean warming are mostly in the regions where there are considerable shortwave reflective clouds, including the marine stratocumulus regions. A recent study has shown that breakup of the marine stratocumulus clouds could speed up ocean warming (Schneider et al. 2019). We added a further explanation on how local marine stratocumulus can change global mean SST (**L192-197; L202-204**).

Schneider, T., Kaul, C.M. & Pressel, K.G. Possible climate transitions from breakup of stratocumulus decks under greenhouse warming. *Nat. Geosci.* 12, 163–167 (2019).

<https://doi.org/10.1038/s41561-019-0310-1>

Fig. R1 Shortwave cloud radiation in boreal summer and its change under warming associated with spread in GMST. (a) Geographical distribution of shortwave ($W m^{-2}$) reflected by cloud in historical simulation of the MME (1956-2005). (b) Surface downward cloud shortwave change regressed on intermodel GMST change ($W m^{-2} K^{-1}$) under RCP8.5 scenario. Dotted regions denote significance at the 5% level.

A few more comments are as follows:

4. Line 173: precipitation heating -> convective heating

Response: Done. (L179)

5. *Line 175: Gill response -> Gill-type response*

Response: Done. (L181)

6. *Line 255: Does the 'absolute SST' means 'area-averaged SST'?*

Response: To make it clearer, "absolute SST" is deleted. (L277-278)

7. *Line 261-263: I think the author should choose either subtracting the mean tropical SST or just using original SST when defining T1 and T2. The low correlation coefficient between PC2 and T2 when using original SST cannot justify the methodology and it gives an impression that the method used in this study is arbitrary.*

Response: Following your suggestion, we choose subtracting the mean tropical SST to define both T1 and T2 (L286-288). The related Fig. 2, Fig. 3, Fig. 4, Supplementary Fig. 5 are updated.

REVIEWERS' COMMENTS round three:

Reviewer #3 (Remarks to the Author):

Methodology and mechanisms are now clearer and the manuscript is now much improved and easy to follow. I would recommend to accept the paper.

REVIEWERS' COMMENTS:

Reviewer #3 (Remarks to the Author): Methodology and mechanisms are now clearer and the manuscript is now much improved and easy to follow. I would recommend to accept the paper.

Response: *Thank you very much for your approval for our revision.*